# Insecticidal Activity of Aqueous Extracts of Plant Origin on *Mahanarva spectabilis* (Distant, 1909) (Hemiptera: Cercopidae)

Vinícius Ferraz Nascimento [1], Alexander Machado Auad [2,*], Tiago Teixeira de Resende [2], Amanda Jdenaina Mendoza Visconde [3] and Marcelle Leandro Dias [1]

1 Department of Biodiversity and Nature Conservation, Federal University of Juiz de Fora, Juiz de Fora 36036900, Minas Gerais, Brazil; viniciusferraz.agro@outlook.com (V.F.N.); marcelle.leandrodias@gmail.com (M.L.D.)

2 Laboratory of Entomology, Embrapa Dairy Cattle, Juiz de Fora 36038330, Minas Gerais, Brazil; tiago.resende@embrapa.br

3 Campus Arnaldo Janssen, UniAcademia University Center, Juiz de Fora 36030776, Minas Gerais, Brazil; amandamendoza.17@hotmail.com

* Correspondence: alexander.auad@embrapa.br

**Abstract:** The spittlebug *Mahanarva spectabilis* (Distant, 1909) is considered the main and most limiting pest of elephant grass (*Pennisetum purpureum* Schum.), and has caused economic losses to Brazilian farmers. In this study, we evaluated the insecticidal effects of aqueous extracts of garlic, rue, cinnamon, lemongrass, clove, star anise, eucalyptus, tobacco, and thyme on *M. spectabilis* nymphs. The results indicated that the tobacco extract was the most effective among all extracts, as it reached a mortality of 76%. The next most effective was star anise extract and cinnamon extract was the least effective. In the second stage of tests, we evaluated the insecticidal potential of five concentrations of tobacco extract. The values of LD50 and LD90 calculated for aqueous tobacco extract were, respectively, 11.5% and 33.3% 24 h after application, and 9.2% and 29.5% 48 h after application. We also evaluated different methods of extraction, through comparative tests, and the methods of infusion and decoction proved to be equivalent to those of the method of extraction by ultrasound. We conclude that among the nine aqueous plant extracts tested, tobacco extract was the only one recommended against *M. spectabilis*, as it showed insecticidal activity, with better results at a concentration of 25%, and with extraction by infusion and decoction.

**Keywords:** spittlebug; mortality; plant active ingredients; extraction methods; agroecology; botanical insecticides; pest control

## 1. Introduction

Spittlebugs (Hemiptera: Cercopidae) are pests of great economic importance in Tropical America, they attack sugar cane fields, pastures and weeds and cause damage to the dairy and cattle farming. Losses of up to USD 2.1 billion per year are estimated due to the action of spittlebugs worldwide [1]. The greatest losses are caused by the adults of this insect pest, which when feeding on the sap of the plant, injects toxins that cause burning of the leaves [2]. Additionally, this pest, especially in its young stage, sucks the xylem content in the roots, interfering with the transport of water and nutrients to the shoot of the plant [3].

Among the species, *Mahanarva spectabilis* (Distant, 1909) (Hemiptera: Cercopidae) is the main and most limiting pest in the elephant grass (*Pennisetum purpureum* Schum [4]. To avoid losses, research has been carried out in search of strategies that are economically and ecologically viable for producers. Embrapa Gado de Leite (Embrapa Dairy Cattle), a Brazilian agricultural research agency, has been constantly publishing results of research on resistant and tolerant plants [5,6] in addition to making partnerships with researchers from Brazilian public universities, for research with biological [7] and chemical [8] control



and on pasture diversification [9]. It should be noted that all of these techniques are part of the integration of control methods to combat this pest.

Among the mentioned strategies, chemical control with synthetic molecules increases the costs of control, mainly affecting family farmers. In Brazil, for example, according to the 2017 Census of Agriculture, conducted by the Brazilian Institute of Geography and Statistics (IBGE), there are about 1.17 million establishments producing milk, with about 60% of the milk produced in the country coming from properties that fall into the category of family farming, described in Law no. 11,326/2006 [10,11]. From the social point of view, the creation of alternative forms of pest control, of low cost and easy acquisition, are fundamental measures for the maintenance of family farming [12].

In another perspective, pest control with synthetic molecules is responsible for generalized environmental pollution, inflicting negative effects on non-target organisms and human health, in addition to being counterproductive, as it leads to the development of resistance in insects [13,14]. In this sense, considering social and environmental aspects, botanical insecticides are alternatives to replace and/or reduce the application of synthetic insecticides, since their correct use can prove to be ecological, cheap and versatile [15].

The use of botanical insecticides is widespread in organic and family farming, especially in low-income countries. Based mainly on the traditional knowledge and secular empirical observations, farmers use homemade extracts from numerous plant species, which have active ingredients with insecticidal and repellency activity, to combat pests in different cultures [16,17]. This is only possible because plants produce a number of bioactive compounds that act as physical and chemical defenses against herbivorous organisms [18]. Volatile organic compounds (VOCs) modulate the plant's relationship to predators, natural enemies and pollinators, inducing responses of attraction and repellence [19]. For example, the species *Sitophilus granarius* (L.) (Coleoptera: Curculionidae) and *Tribolium confusum* Jacquelin du Val (Coleoptera: Tenebrionidae), stored grain pests, respond to blends of cereal kernels and plant volatiles [20,21]. *Mahanarva spectabilis* is also able to detect and respond to the presence of VOCs [22,23]. In addition to olfactory responses, some plant compounds can be toxic, cause sterility, modify development, and reduce insect feeding [24]. In plants, these compounds are accumulated in small proportions in plant tissues, and from them it is possible to make powders, botanical extracts and essential oils that can be used as insecticides, repellents and attractives in agriculture [25].

Garlic (*Allium sativum* L.) is a plant in the Alliaceae family, which presents significant amounts of bioactive organosulfur compounds, mainly concentrated in the bulbs. Its botanical potential against the pests has been proven, for example, on *Aedes aegypti* (Diptera: Culicidae) and *Tribolium castaneum* (Herbst) (Coleoptera: Tenebrionidae) [15,26]. Tobacco (*Nicotiana tabacum* L.), which is widely used in agriculture in the form of an extract, owes its insecticidal properties to nicotine and other alkaloids [17]. Researchers have demonstrated its efficiency against hemiptera, coleoptera, and lepidopterans [27–29].

The essential oil (OE) of thyme (*Thymus vulgaris* L.) is rich in thymol and carvacrol, substances tested and described in the literature as potent insecticides and acaricides [8,30] (E)-cinamaldehyde is the main aromatic component present in the OE of cinnamon (*Cinnamomum verum* J. Presl), that has demonstrated high insecticidal potential against two species of dipterans [31]. The most abundant active compound in star anise (*Illicium verum* Hook f.) is trans-anethole, whose insecticidal effect has already been utilized on *Sitophilus zeamais* Motschulsky (Coleoptera: Curculionidae) and *M. spectabilis*, with promising results [8,32]. The clove (*Syzygium aromaticum* L.) has the greatest bioactive constituents, such as Eugenol and E-caryophyllene, the former being a compound tested by many researchers against pests, with good results [8,33,34]. Eucalyptus (*Eucalyptus globulus* Labill) has its insecticidal properties due to the presence of the component 1,8-cineole in its essential oil [35,36]. The lemongrass OE (*Cymbopogon citratus* Stapf) has shown efficiency as a repellent and/or insecticide in different studies [37–39]. The rue (*Ruta graveolens* L.), in the form of powder, OE, and aqueous extract, has active ingredients that provoke its insecticidal effects [40–42].

Few published studies have tested the insecticidal effects of the compounds of plant origin against the spittlebugs of the genus *Mahanarva*. Garcia et al. [43] proved the potential of neem-based (*Azadiractha indica* A. Juss) products and extracts, for the control of *Mahanarva fimbriolata* (Stål) (Hemiptera: Cercopidae). Likewise, Pistori et al. [44] demonstrated that the aqueous extract of *Anacardium humile* St. Hill caused significant reductions in the nymphal survival rate of *M. fimbriolata*. Dias et al. [8] tested the insecticidal potential of different compounds of plant origin, acquired in their standard chemical form, on the spittlebug *M. spectabilis*. The researchers obtained expressive results in the control of nymphs and adults with the compound transanethole, which is also present in star anise. These works demonstrate that the extracts and compounds of botanical origin can be alternatives to the control of spittlebugs with synthetic chemical pesticides and that more research is needed, which can take into account the factors as plants, concentrations, and forms of extraction that could serve the small producers.

On top of this, the market for botanical insecticides is booming and is expected to continue to grow by 2025 [45]. With an increasing demand for safe and contaminant-free products, botanical insecticides can reach various agricultural sectors, particularly in the management of pests in crops of high added value, such as fruit and vegetables, and in crops that directly affect the feeding of meat-producing animals, such as pastures and grains [45,46].

In the present study, the contact toxicity of aqueous extracts of *A. sativum*, *R. graveolens*, *C. verum*, *C. citratus*, *S. aromaticum*, *I. verum*, *E. globulus*, *N. tabacum* and *T. vulgaris* against nymphs of *M. spectabilis* was tested to explore their insecticide potential. We also tested five concentrations and three forms of extraction for the aqueous extract of *N. tabacum*, on nymphs of *M. spectabilis*.

## 2. Material and Methods

### 2.1. Botanical Material

For the formation of elephant grass seedlings (*Pennisetum purpureum* Schum), stakes of approximately 10 cm, with a single node, were obtained from plants in the experimental field of Embrapa Dairy Cattle in Coronel Pacheco, MG, Brazil. The cuttings were propagated in plastic pots (500 mL) that contained the substrate (soil/manure in the proportion 1:1), to form the seedlings. The seedlings were kept in a greenhouse for about 60 days, until they were used in the experiments as feeding substrates for the nymphs.

Seedlings of lemongrass, rue and thyme were planted and irrigated daily in plastic pots of 1 L and stored in the greenhouse at the Embrapa Dairy Cattle for 3 months. These seedlings and the dried parts of cinnamon, cloves, star anise and tobacco were purchased from the traders of the Municipal Market of Juiz de Fora, MG, Brazil. Organic garlic was purchased from the street fair at Juiz de Fora. Eucalyptus leaves were collected from the 12-year-old trees planted in Maripá de Minas, MG, Brazil. The parts used in the preparation of the extracts and their active ingredients are listed in Table 1.

### 2.2. Processing of Botanical Material

To obtain the botanical extracts, the selected parts of each plant (Table 1) were washed with distilled water and then distributed on the sheets of paper in metal trays. Consequently, they were placed in an oven of forced ventilation (model FD115—BINDER, Tuttlingen, Germany) at an average temperature of 40 °C for 72 h.

The kiln-dried materials were milled in a basic analytical mill (model A11—IKA, Staufen, Germany), and placed in beckers. After this procedure, the plant extracts were prepared by mixing the milled material (from 3 g to 25 g, depending on the bioassay), with distilled water (100 mL), for 10 min in an ultrasound cleaner (model E60H—Elma, Singen, Germany). The resulting solutions were filtered using a voile fabric, which gave rise to aqueous extracts. The extracts were then stored in the bottles of amber glass, protected from light, at a temperature of 20 ± 4 °C for up to 24 h on average before bioassays.

**Table 1.** List of the species of plants and their parts used to make the aqueous extracts.

| Plant Species | Family | Parts Used | Active Ingredients | References |
|---|---|---|---|---|
| Garlic (*Allium sativum* L.). | Alliaceae | Bulbs | Methyl allyl disulfide and Diallyl trisulfide | Huang et al. [47,48] |
| Tobacco (*Nicotiana tabacum* L.) | Solanaceae | Leaves | Nicotine | Dougoud et al. [17] |
| Lemongrass (*Cymbopogon citratus* Stapf.) | Poaceae | Leaves | Geranial and Neral | Olivero-Verbel et al. [49] |
| Eucalyptus (*Eucalyptus globulus* Labill.) | Myrtaceae | Leaves | 1,8-cineole | Mossi et al. [35] |
| Rue (*Ruta graveolens* L.) | Rutaceae | Leaves | 2-Nonanone and 2-Undecanone | Orlanda and Nascimento [50] |
| Thyme (*Thymus vulgaris* L.) | Lamiaceae | Leaves | Thymol and Carvacrol | Park et al. [51] |
| Cinnamon (*Cinnamomum verum* J. Presl) | Lauraceae | Bark | Cinnamaldehyde | Benelli et al. [31] |
| Star anise (*Illicium verum* Hook.f) | Illiciaceae | Fruits | Trans-anethole | Wei et al. [32] |
| Clove (*Syzygium aromaticum*, L.) | Myrtaceae | Flower bud | Eugenol and E-caryophyllene | Zeng et al. [33] |

*2.3. Insects*

For the realization of the bioassays, nymphs of *M. spectabilis* of fourth and/or fifth instar, of unidentified sex, were collected from the elephant grass cv. Kurumi at the Embrapa Dairy Cattle Experimental Field and taken to the Laboratory of Entomology. In the collection procedure, the nymphs were removed from the base of the plant with the help of a brush, placed in beakers containing roots, and kept in plants in the laboratory until use in the experiment. The collection of approximately 1000 nymphs was performed on different dates, in the mornings before each bioassay.

*2.4. Assessment of Insecticidal Activity*

2.4.1. Comparative Bioassays between Extracts

For the development of the bioassay, the methodology of Dias et al. [8] was used. Two bioassays were carried out with different concentrations, 3% and 20%, to compare the extracts against their insecticidal potential on the *M. spectabilis* nymphs. In the tests, nine extracts (Table 1), positive controls (Tiametoxam 141 g/L + Lambda-cyhalotrin 106 g/L), and negative controls (distilled water) were used, totaling 11 treatments with 10 repetitions.

In each repetition, 10 nymphs were distributed over Petri dishes, and each nymph received 10 µL of the solution on its back, through a micropipette (model 0.5–10 uL V3-Plus—Ulster Scientific, New Paltz, NY, USA). Consequently, the insects were transferred to the plastic pots (500 mL) with elephant grass plants, arranged in random blocks. The plants used had their roots previously exposed to facilitate nymphal feeding. The cups were wrapped in the bags of "voil" fabric to prevent the nymphs from escaping. The experiment was maintained in a Walk-In Climate Chamber (Eletrolab, São Paulo, Brazil) at 25 ± 2 °C, at the humidity of 70% ± 10%, and with the photophase of 12 h, during the tests. The insecticidal activity of each extract was evaluated after 24 and 48 h after its application, and subsequently, the number of alive and killed nymphs by the insecticidal action was counted.

2.4.2. Comparative Bioassay between Tobacco Concentrations

The tobacco extract showed satisfactory results in previous bioassays. Based on this information, it was decided to continue the tests with a bioassay comparing different concentrations of the tobacco extract.

Five concentrations (5%, 10%, 15%, 20% and 25%) of aqueous tobacco extract were prepared according to item 2.2. In the bioassay, each concentration was considered as a treatment, and distilled water was used as a control. Ten repetitions were performed. The test methodology was the same as previously described in the comparative bioassay between the extracts. The insecticidal activity of each concentration of the extract was evaluated after 24 and 48 h of application, and the number of the nymphs alive and killed by the insecticidal action was counted.

### 2.4.3. Comparative Bioassay between Extraction Methods

In this bioassay, four forms of extraction were tested and compared in relation to the efficiency of extraction in tobacco, comparing the results through the nymph survival of *M. spectabilis*; they are Static Maceration, Infusion, Decoction and Ultrasound Assisted Extraction (UAE). The UAE was the basis for comparisons because it is the methodology adopted in the other bioassays of the present research.

The method of extraction by static maceration consisted of placing the pressed and shredded tobacco leaves in contact with distilled water for 24 h, and then separating the liquid part from the solid with the help of filtration using the voile fabric. In the decoction process, the pressed and shredded sheets of tobacco were placed in a container with distilled water. The container was taken to a magnetic stirrer with temperature control (model TE-0852—Tecnal, Piracicaba, Brazil) and kept until it boiled (~15 min). At the end of the process, the extract was filtered through the voile fabric. In the infusion process, the tobacco was dried in an oven of forced ventilation (model FD115—BINDER) at an average temperature of 40 °C, for 72 h, and milled in a basic analytical mill (model A11—IKA). The resulting powder was placed in contact with the boiling distilled water (100 °C) and left until it was cooled (~20 min). The extract was filtered through the voile fabric.

In the bioassay, all the extracts were prepared in the concentration of 25%, and distilled water was used as a control, totaling in five treatments. Ten repetitions were performed. In each repetition, 10 nymphs were distributed on the Petri dishes. The test and evaluation methodology was the same as the one previously described in the comparative bioassay between the extracts.

### 2.5. Statistical Analysis

Mortality values were transformed in arcsin sqrt (x + 1) and subjected to an analysis of variance (ANOVA) and means were compared by using Scott Knott test ($p < 0.05$), using the free software Sisvar, version 5,6, build 90 [52]. To determine the toxicity regression, LC50 and LC90, the Probit analysis was used, with 95% confidence intervals, using the Probit (LeOra Software POLO-Plus 1.0) [53].

## 3. Results

### 3.1. Comparative Bioassays between Extracts

A significant difference was found in the mortality of nymphs (F = 199.98; $p < 0.0001$) between the negative control and the aqueous extracts of tobacco, rue and cloves, at a concentration of 3%, 24 h after application. Although they differ significantly, the mortality caused the was less than 9%. The insecticide used (positive control) promoted 100% mortality of nymphs of *M. spectabilis* (Figure 1). In the evaluation after 48 h, only the tobacco extract differed significantly (F = 108.39; $p < 0.0001$) from the negative control, with a mortality of 26% (Figure 1).

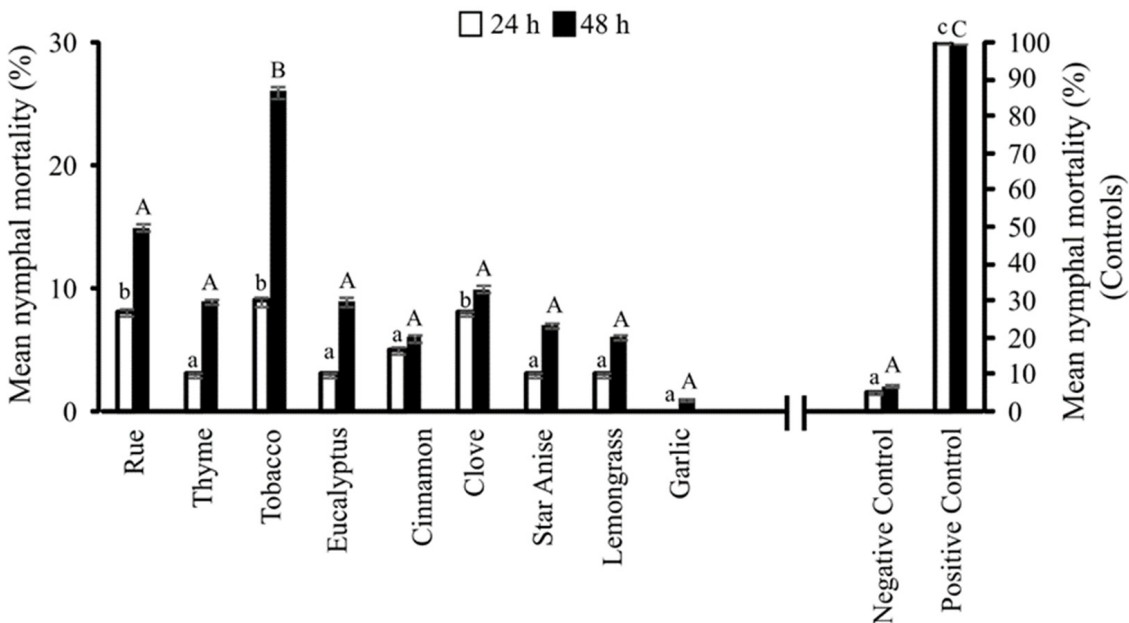

**Figure 1.** Insecticidal activity of botanical extracts (concentration of 3%) against *Mahanarva spectabilis* nymphs after 24 and 48 h of application. Different letters represent significant differences between the treatments at 24 h (lower case letter) and 48 h (capital letters) by the Scott Knott test ($p < 0.05$).

At a concentration of 20%, the tobacco extract promoted significant lethal effects in the nymphs of *M. spectabilis*, after 24 h (F = 150.78; $p < 0.0001$) and 48 h (F = 140.36; $p < 00001$) of application, in comparison with the negative control and the other extracts. The mortality values caused by tobacco extract were 68% and 76% in the evaluations performed after 24 and 48 h, respectively (Figure 2). It could also be observed that in the evaluation carried out at 48 h of application, in addition to the tobacco extract, the extracts of star anise and lemongrass differed significantly from the negative control, but were 4.2 times lower in terms of deadliness to nymphs than the tobacco extract (Figure 2).

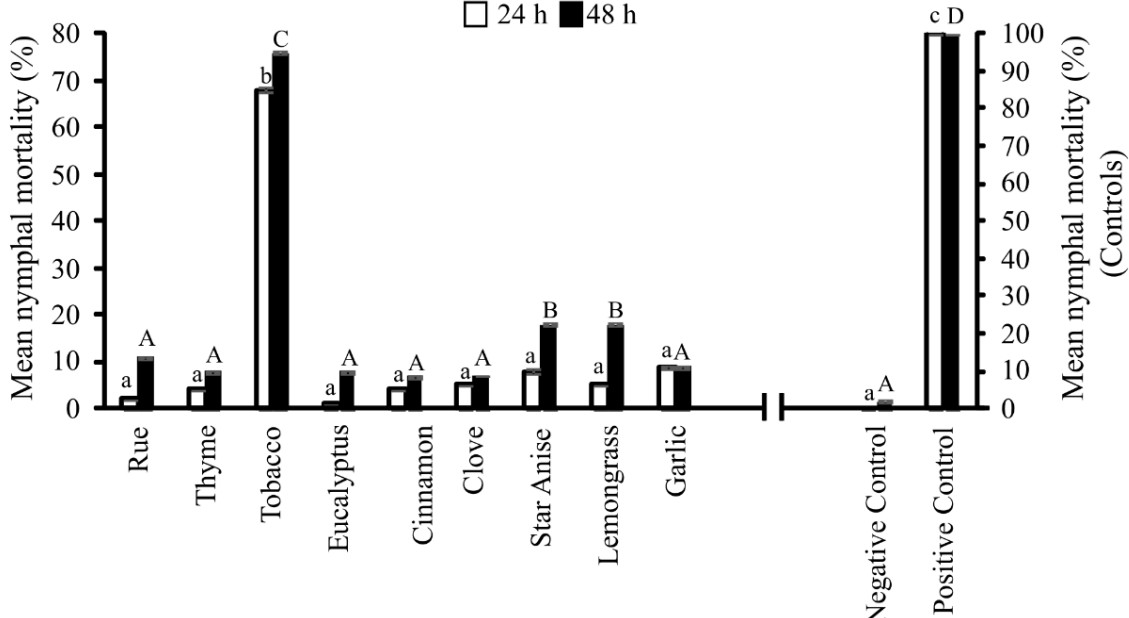

**Figure 2.** Insecticidal activity of botanical extracts (concentration of 20%) against *Mahanarva spectabilis* nymphs after 24 and 48 h of application. Different letters represent significant differences between the treatments at 24 h (lower case letter) and 48 h (capital letters) by the Scott Knott test ($p < 0.05$).

### 3.2. Comparative Bioassay between Tobacco Concentrations

A significant nymphal decrease in *M. spectabilis* was observed due to the increase in the concentration of tobacco extract after 24 h (F = 173.73; $p < 0.0001$) and 48 h (F = 211.05; $p < 0.001$) (Figure 3). A 95% mortality of *M. spectabilis* was found when applied to the tobacco extract at a concentration of 25%. Intermediate mortalities (64 and 79%) were obtained at concentrations of 15 and 20% of tobacco extracts after 48 h of application. In the other concentrations, the extracts differed significantly from the negative control, but reached a mortality equal to or less than 50%. The values of LD50 and LD90 calculated for the aqueous tobacco extract were, respectively, 11.5% and 33.3% in 24 h, and 9.2% and 29.5% 48 h after application.

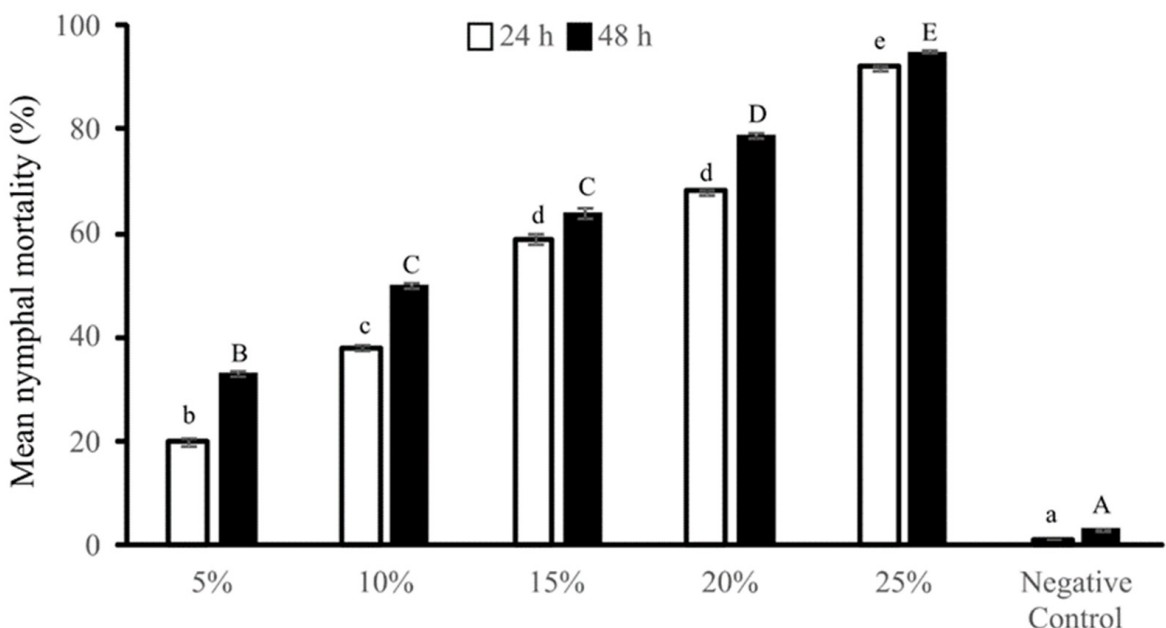

**Figure 3.** Insecticidal activity of different concentrations of tobacco extracts against *Mahanarva spectabilis* nymphs after 24 and 48 h of application. Different letters represent significant differences between the treatments at 24 h (lower case letter) and 48 h (capital letters) by the Scott Knott test ($p < 0.05$).

### 3.3. Comparative Bioassay between Extraction Methods

In the first evaluation, performed 24 h after the application of the tobacco extracts, the mortality of *M. spectabilis* nymphs was significantly higher (F = 52.57; $p < 0.0001$) in relation to the negative control in all treatments. The mortalities ranged from 63 to 91% depending on the extraction method. Static maceration, UAE and decoction did not differ significantly from each other and promoted mortality lower than the infusion method, which was the most effective in promoting the nymph mortality of *M. spectabilis* (Figure 4).

After 48 h of application, all extracts promoted significant differences (F = 116.34; $p < 0.0001$) in the mortality of insect pest nymphs, compared to the negative control treatment. The mortality ranged from 73 to 96%. In evaluating treatments, UAE and decoction corresponded significantly to the infusion that was more effective in 24 h, but differed from the static maceration treatment that was less effective (Figure 4).

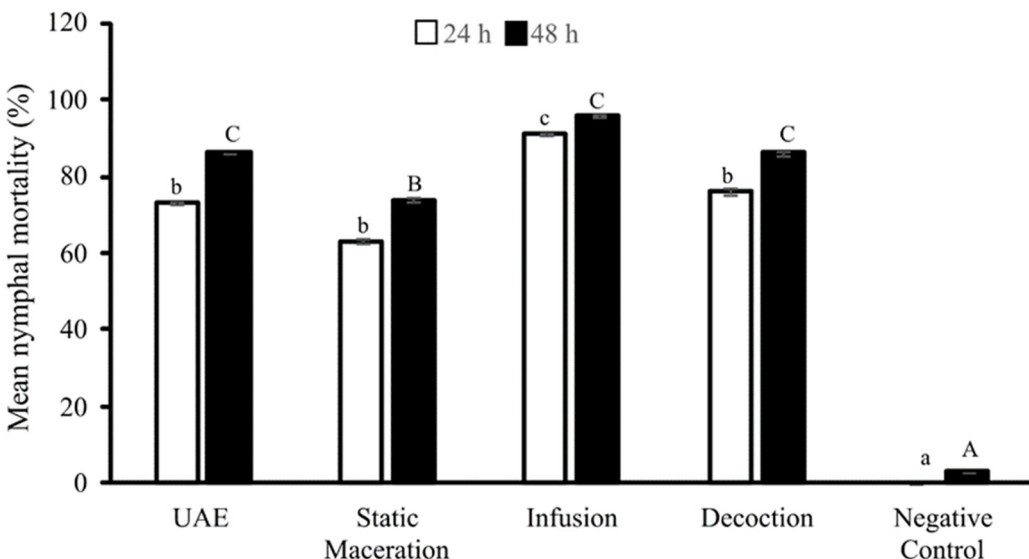

**Figure 4.** Insecticidal activity of tobacco extracts produced by different extraction methods against *Mahanarva spectabilis* nymphs after 24 and 48 h of application. Different letters represent significant differences between the treatments at 24 h (lower case letter) and 48 h (capital letters) by the Scott Knott test ($p < 0.05$).

## 4. Discussion

In nature, many chemical compounds from the secondary metabolism of plant species have properties that inhibit the action of herbivorous insects [54]. The action can occur in terms of mortality, inhibition of food, reduction in food consumption, delay in the development, deformations and, sterility [55]. Thus, many plant species provide potential sources of compounds for the control of insect pests, which can be obtained by making the extracts in different solvents. These extracts act as botanical insecticides and are more sustainable alternatives to pest management than synthetic insecticides [56,57].

However, it is a mistake to consider all botanical insecticides as harmless products, because the toxicity of a molecule is due to its chemical structure and not its origin [58]. Some plant compounds, such as nicotine, azadirachtin, rotenone and strychnine, can be acutely toxic to mammals, natural enemies and pollinators, in their pure form [59,60]. However, the risks associated with these compounds are largely mitigated using plant extracts, where the concentrations of the compounds are low [61]. Therefore, it is assumed that each insecticide, synthetic, biological or botanical, must be tested for environmental persistence and toxicity to non-target organisms.

This research revealed the effect of aqueous extracts obtained from nine different plants, on the nymphs of the spittlebug *Mahanarva spectabilis*. The low mortality observed for extracts of garlic, rue, cinnamon, lemongrass, clove, eucalyptus and thyme may be associated with differences in the concentration of active ingredients in extracts and plants, and also with low residual effects [62]. Another association that can be made is that, according to Dias et al. [8] the foam produced by the nymphs, after applying the treatments, can act in the partial elimination of the irritant. At a concentration of 20%, treatments with aqueous extracts of lemongrass and star anise differed significantly from the negative control, but registered low mortality. Other authors have obtained expressive results with the extracts of lemongrass and star anise. Karunamoorthi and Ilango [63] used the methanolic extract of lemongrass and observed a high larvicidal efficacy against *Anopheles arabiensis* Patton (Diptera: Culicidae), which is among the main vectors of malaria in Africa. Additionally, in the work of Zhou et al. [64], the star anise extract, in ethyl alcohol, ethyl acetate and petroleum ether solvents, was highly toxic to the adults of *Myzus persicae* (Sulzer) (Hemiptera: Aphididae). Thus, considering the results obtained, it is proposed that the difference in toxicity observed between the present research and that of other authors

may have occurred due to the particularities of the species used in each study, such as physiology and resistance mechanisms.

In Brazil, the National Health Surveillance Agency (ANVISA) considers tests with insecticides to be satisfactory when the mortality of the treatment reaches an average value of 90% (±10%) in relation to the control [65]. None of the extracts observed in the first bioassay promoted satisfactory mortality results for ANVISA parameters; however, the aqueous tobacco extract showed results close to the satisfactory value. Other studies have obtained satisfactory results with the tobacco extract, as in that of Sarker and Lim [66] in which the methanolic tobacco extract reduced the survival of the first instar and adult caterpillars of *Grapholita molesta* (Busck) (Lepidoptera: Tortricidae) among the 32 plant extracts tested. One explanation for the fact that this extract stands out as the most efficient is due to the presence of compounds with insecticidal properties.

Recently, the GC/MS analysis performed by Kanmani et al. [67] determined that the major compounds present in hexane, petroleum ether, dichloromethane, chloroform, ethyl acetate, acetone, methanol and aqueous tobacco leaf extracts are nicotine, nicotinonitrile, nornicotine, nicotinic acid, neonicotine, cotinine, indole, farnesol, sclareol, 9,12-octadecadienoic acid, squalene, palmitic acid, and 15-tetracosenoic acid methyl ester. Among the compounds, the nicotine alkaloid is the main active substance present in tobacco plants, and its extraction is possible using water as a solvent [68,69]. In insects, it causes nerve impulses that lead to the hyperexcitation of the insect's nervous system and, consequently, its death [70]. In general, it is known that tobacco extract acts as a contact insecticide, which has fast action and degrades in the environment, it has low phytotoxicity, it not very harmful to the soil and is low cost [70,71]. Thus, based on the preliminary results, it was decided to carry out a trial to compare the effects of different concentrations of tobacco extract on *M. spectabilis*.

Defining the best concentration is important, which should be neither high enough to be wasted, causing costs and environmental problems, nor scarce, to be inefficient to lead the insects to develop resistance [72]. In other words, the appropriate concentration allows us to determine the intensity of the field treatment after the preliminary screening in the laboratory. The results of this bioassay confirmed that all tested concentrations of aqueous tobacco extract showed significant insecticidal activity against nymphs of *M. spectabilis*. Similar observations about the effect of *N. tabacum* extract on other insect species have been reported. Rizvi et al. [73] showed that the ether tobacco extract significantly reduced the infestation levels of *Trichoplusia binotalis* Hiibner (Lepidoptera: Noctuidae) in cabbage plants (*Brassica oleracea* L.) compared to the controls. The researchers Lokesh et al. [74] reported 100% mortality of the adults of *Cylas formicarius* Fabricius (Coleoptera: Brentidae) after 72 h, which were submitted to the extract of chloroform and acetone from the tobacco leaves. Natural tobacco-based insecticides have been used by man since the 18th century, but these insecticides were gradually replaced in the last century with the synthetic ones, with substances that are highly aggressive to man and to the environment [75,76]. However, due to the health and environmental problems generated by the large-scale use of synthetic insecticides and the market's interest in more sustainable products, there is a growing interest of the farmers in the alternative products that can be used for pest management [16,48].

However, for the adoption of alternative technologies such as plant extracts, it is necessary that their preparation, extraction processes, be easy and viable for farmers on their properties. The methods of extracting hot compounds, such as decoction and infusion, consist of extracting active ingredients from the plants. Since plants are degraded by the combined action of water and heat, the methods become simple, fast, and feasible for the farmers [77]. One of the first mentions of insecticides and forms of extraction, in 1763, already reports the use of the infusion of tobacco leaves for the control of lice [78]. Currently, methods of extracting plant compounds by infusion and decoction are still widely used; Cuevas-Salgado et al. [79] obtained significant mortality rates of the eggs and caterpillars of *Leptophobia aripa* Boisduval (Lepidoptera: Pieridae) by using botanical tobacco infusion. In their research, 48 h after the application of the extracts, the methods of

infusion and decoction statically matched with the UAE method (laboratory control). The infusion method was more effective than the UAE. Therefore, it is the most recommended method for the extraction of the bioactive compounds of tobacco by the farmers. The results reported in this work open the possibility of further investigations into the efficacy of the tobacco extract, and its insecticidal properties on *M. spectabilis*, in field conditions, and for the future recommendations in the programs of integrated pest management on small farms, since it showed the highest insecticidal bioactivity at the concentration of 25%, with its extraction using infusion and decoction which are easy and viable for famers to carry out on their properties.

**Author Contributions:** Conceptualization, V.F.N. and A.M.A.; methodology, V.F.N. and A.M.A.; investigation, all authors; resources, A.M.A.; data curation, V.F.N. and A.M.A.; writing—original draft preparation, V.F.N. and A.M.A.; writing—review and editing, all authors; visualization, all authors; supervision, A.M.A. All authors have read and agreed to the published version of the manuscript.

**Funding:** This research was funded by the Coordenação de Aperfeiçoamento de Pessoal de Nível Superior (CAPES), Brazil (Finance Code 001); Conselho Nacional de Desenvolvimento Científico e Tecnológico (CNPq), Brazil (Finance Code 304281/2019-0); Fundação de Amparo à Pesquisa do Estado de Minas Gerais (FAPEMIG, Brazil (Finance Code CAG APQ-00732-18).

**Institutional Review Board Statement:** Not applicable.

**Informed Consent Statement:** Not applicable.

**Data Availability Statement:** The datasets generated and/or evaluated during the current study are available from the first author when request.

**Acknowledgments:** The authors express their gratitude to Embrapa Dairy Cattle (Embrapa Gado de Leite), Brazil; Coordenação de Aperfeiçoamento de Pessoal de Nível Superior (CAPES), Brazil. The Federal University of Juiz de Fora (Universidade Federal de Juiz de For a—UFJF), Brazil; the Conselho Nacional de Desenvolvimento Científico e Tecnológico (CNPq), Brazil (Finance Code 304281/2019-0) and Fundação de Amparo à Pesquisa do Estado de Minas Gerais (FAPEMIG, Brazil (Finance Code CAG APQ-00732-18).

**Conflicts of Interest:** The authors declare no conflict of interest.

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
