# Peer review of "Insecticidal Activity of Aqueous Extracts of Plant Origin on Mahanarva spectabilis (Distant, 1909) (Hemiptera: Cercopidae)"

_agronomy, doi:10.3390/agronomy12040947_

Round 1

Reviewer 1 Report

Dear authors,

your study brings results which could be applied in pest control. Nevertheless, I have the following suggestions for improvement. One of the most important concern is how data are analysed and presented.

Abstract:

It would be good to add at lest one plant extract which has highest efficacy in addition to tobacco and mention the least efficacy extract, too. LC50 value should calculated from concentration-response experiments.

L27 insecticidal bioactivity > insecticidal activity

Keywords:

suggested to add "botanical insecticides" and "pest control"

Introduction:

It is almost well written covering the most important knowledge on the studied topic from general to specific. I suggest to consult review on botanical pesticides by Pavela (doi 10.17221/31/2016-PPS)

L 33 - remove coma in "... pastures and, ..."

L 46 - with biological control [7] chemical [8] should read "with biological [7] and chemical [8] control"

L 61-62 this requires linguistic correction "reduce synthetic insecticides, in
chemical control, ..." I suggest to write "reduce application of synthetic insecticides" and delete chemical control

L 75  and onwards - numbers of references should be probably in the form [19-21]

L 77 - reference to paper of doi 10.3390/molecules22111873 demonstrating strong acarological effect can be added

Material and Methods:

L 12-123 last sentence is confusing as all plants origin is described above, moreover "vegetables" is probably wrong term used in this context

L 132 but in bioassays below only 3 and 20% concentrations were used, this should be clarified what concentration was for which bioassay or do not specify exact numbers here but in particular bioassay or just to write "between 3 and 25 g depending on bioassay"

L 136 for how long (in average) and at what temperature?

L 147 please correct citation format [8]

L 178-180 since UAE method is described earlier it would be good to use this procedure name (UAE) in section 2.2 (around line 133)

L 200 correct spelling Abbott

L 201 as regards mortality data they are not normally distributed so either transformation is needed (arcsin sqrt) and since there are two measurements (at 24 and 48 hours), repeated measures ANOVA were time is the second factor should be applied. However, instead of transformation and ANOVA, generalized linear models with binomial function are recommended

L 202-203 for dose-response data standard procedure is to use probit analysis (special type of regression), I highly recommend to use it and calculate LC50 and LC90

L 212 it is better to be consistent in efficacy values - I suggest to keep % of mortality instead of survival

Figures 1 and 2 There is no need to show both survival and mortality. Only mortality data are interesting, use raw (not corrected for mortality in control using Abbott - corrected values can be written in text, alternatively, show in graphs corrected values without control column and write in the text what was was mortality in negative control) and if positive control was 100% use broken y-axis or write this value only in text so that y-axis max is about 20-30 and differences between treatments are more obvious. You can consider to merge either time series data (24 and 48 h) or concentrations in one graph

L 232-233, Fig. 3 as recommended above, probit analysis would be more appropriate

Figure 4 is redundant when concentration-mortality curves are shown in Fig. 3

Discussion:

I suggest to address also potential health and environmental risks of botanic insecticides

L 293 I would expect that very important is which solvent is used, too. Particularly in case of active substances which are difficult to extract using water and thus alcoholic extracts might show better results

L 350 spectabilis (lowercase letter s)

Conclusions:

This paragraph could be more elaborated, single-sentence paragraph looks too short.

Author Response

Responses to the reviewer’s comments

 We sincerely appreciate all valuable comments and suggestions, which helped us to improve the quality of the manuscript. Our responses to your and the Reviewer's comments are described below in a point-to-point manner.

Reviewer 1 comments

>>1. Your study brings results which could be applied in pest control. Nevertheless, I have the following suggestions for improvement. One of the most important concern is how data are analysed and presented

#Response: Dear reviewer, we are grateful for your feedback. We made some changes in the way the data are presented, taking into account the suggestions.

>>2. It would be good to add at lest one plant extract which has highest efficacy in addition to tobacco and mention the least efficacy extract, too. LC50 value should calculated from concentration-response experiments.

#Response: We made the suggested changes, but adapted the abstract to the limit of 200 words described in the guide for authors.

>>3. L27 insecticidal bioactivity > insecticidal activity

#Response: Done.

>>4. Suggested to add "botanical insecticides" and "pest control"

#Response: Done.

>>5. It is almost well written covering the most important knowledge on the studied topic from general to specific. I suggest to consult review on botanical pesticides by Pavela (doi 10.17221/31/2016-PPS)

#Response: We consulted Pavela’s review, and added some information to the introduction text.

>>6. L 33 - remove coma in "... pastures and, ..."

#Response: Done.

>>7. L 46 - with biological control [7] chemical [8] should read "with biological [7] and chemical [8] control"

#Response: Done.

>>8. L 61-62 this requires linguistic correction "reduce synthetic insecticides, in chemical control, ..." I suggest to write "reduce application of synthetic insecticides" and delete chemical control.

#Response: Thank you, we corrected according to your suggestion.

>>9. L 75  and onwards - numbers of references should be probably in the form [19-21]

#Response: Done.

>>10. L 12-123 last sentence is confusing as all plants origin is described above, moreover "vegetables" is probably wrong term used in this context

#Response: We rewrite the sentence.

>>11. L 132 but in bioassays below only 3 and 20% concentrations were used, this should be clarified what concentration was for which bioassay or do not specify exact numbers here but in particular bioassay or just to write "between 3 and 25 g depending on bioassay"

#Response: We heed your suggestion, and rewrite the sentence.

>>12. L 136 for how long (in average) and at what temperature?

 #Response: We add the information to the text.

>>13. L 147 please correct citation format [8]

#Response: Done

>>14. L 178-180 since UAE method is described earlier it would be good to use this procedure name (UAE) in section 2.2 (around line 133).

#Response: We add the information in the correct location.

>>15. L 200 correct spelling Abbott

#Response: Done

>>16. L 201 as regards mortality data they are not normally distributed so either transformation is needed (arcsin sqrt) and since there are two measurements (at 24 and 48 hours), repeated measures ANOVA were time is the second factor should be applied. However, instead of transformation and ANOVA, generalized linear models with binomial function are recommended

#Response: The mortality data were transformed into arcsin sqrt. Although there are two measurements (at 24 and 48 h) they were not applied as a second factor, because the 48 h data were cumulative, so it will always be greater than 24 h. In this new version, generalized linear models with binomial function were not used, but if the reviewer think that must has to be changed, we will do it in the next version.

>>17. L 202-203 for dose-response data standard procedure is to use probit analysis (special type of regression), I highly recommend to use it and calculate LC50 and LC90.

#Response: Done

>>18. L 212 it is better to be consistent in efficacy values - I suggest to keep % of mortality instead of survival.

#Response: We replace the survival data with mortality data.

>>19 Figures 1 and 2 There is no need to show both survival and mortality. Only mortality data are interesting, use raw (not corrected for mortality in control using Abbott - corrected values can be written in text, alternatively, show in graphs corrected values without control column and write in the text what was was mortality in negative control) and if positive control was 100% use broken y-axis or write this value only in text so that y-axis max is about 20-30 and differences between treatments are more obvious. You can consider to merge either time series data (24 and 48 h) or concentrations in one graph

#Response: We appreciate your suggestion and have changed the figures and part of the text.

>> 20 L 232-233, Fig. 3 as recommended above, probit analysis would be more appropriate.

#Response: We kept the figure, but added to the text the values of LC50 and LC90.

>> 21 Figure 4 is redundant when concentration-mortality curves are shown in Fig. 3

#Response: We removed the figure 4 from the manuscript.

>> 22 I suggest to address also potential health and environmental risks of botanic insecticides

#Response: It is very important to address this issue, we appreciate the suggestion, and we add to the text.

>> 23 L 293 I would expect that very important is which solvent is used, too. Particularly in case of active substances which are difficult to extract using water and thus alcoholic extracts might show better results.

#Response: We add the information to the text.

>> 24 L 350 spectabilis (lowercase letter s)

#Response: Done

>> 25 Conclusions: This paragraph could be more elaborated, single-sentence paragraph looks too short.

#Response: We have drawn up a new paragraph.

Sincerely,

Authors

Reviewer 2 Report

Review ID 1660164

Insecticidal Activity of Aqueous Extracts of Plant Origin on 2 Mahanarva spectabilis (Distant, 1909) (Hemiptera: Cercopidae)

               It is obvious that there is a growing interest in environmentally friendly methods of plant protection worldwide. Large number of pesticides that are ineffective and have negative impact on the environment have been reduced. In present time, environmentally friendly methods of plant protection are of importance where natural defense mechanism of plants based on volatile organic compounds may play significant role.

               This is quite well organized manuscript. I found this “ms” interesting and innovative. However, a few questions must be explained more precisely.

Critical review:

  1. Before you start writing about plant extracts it is of importance to mention the natural plant's defence strategy, where volatiles play a crucial role. Some papers below.
  2. The aim of the study is not well explained.
  3. Combine Figures 1 and 2 as well as 4 and 5.
  4. What is a difference between the title of Figure 4 and 5?
  5. Conclussions are not acceptable in present form.

Some other paper to add:

Sitophilus granarius responses to blends of five groups of cereal kernels and one group of plant volatiles

Journal of Stored Products Research 62: 36-39 (2015)

DOI: 10.1016/J.JSPR.2015.03.007

Tribolium confusum responses to blends of cereal kernels and plant volatiles

Journal of Applied Entomology 140, 558–563 (2016)

DOI: 10.1111/JEN.12284

Author Response

Responses to the reviewer’s comments

 We sincerely appreciate all valuable comments and suggestions, which helped us to improve the quality of the manuscript. Our responses to your and the Reviewer's comments are described below in a point-to-point manner.

Reviewer 2 comments

>> 1. Before you start writing about plant extracts it is of importance to mention the natural plant's defence strategy, where volatiles play a crucial role. Some papers below.

#Response: Thanks for the suggestion, we add a paragraph on this theme to the text.

>> 2. The aim of the study is not well explained.

#Response: We try to explain it more easily.

>> 3. Combine Figures 1 and 2 as well as 4 and 5.

#Response: We changed the figures, although we didn’t put them together.

>> 4. What is a difference between the title of Figure 4 and 5?

#Response: We changed the captions of the figures.

>>5. Conclusions are not acceptable in present form.

#Response: We have drawn up a new paragraph.

Sincerely,

Authors

Round 2

Reviewer 1 Report

Dear Authors,

I appreciate changes you made in your revised version of the manuscript. I would recommend few more editing, mostly minor linguistic  changes:

line 76 Species name is usually written in full at the beginning of sentence, i.e. here Mahanarva spectabilis ...

line 160 should read ... an ultrasonic cleaner (model ... ?

line 163 would better read as 20 ± 4 °C for up to 24 hours in average before bioassays. 

lines 251-254 it should be normal (not bold) font and Mahanarva spectabilis should be in italics

lines 276-279, 304-307, 323-326 - the same format corrections as described above are needed

Author Response

Responses to the reviewer’s comments

Reviewer 1 comments

>>1. L76 - Species name is usually written in full at the beginning of sentence, i.e. here Mahanarva spectabilis

#Response: Done.

>>2. L160 - should read ... an ultrasonic cleaner (model ... ?

#Response: Yes, the information has been added to the document.

>>3. L163 - would better read as 20 ± 4 °C for up to 24 hours in average before bioassays.

#Response: Done.

>>4. L251-254 it should be normal (not bold) font and Mahanarva spectabilis should be in italics

#Response: Done.

>>5. Lines 276-279, 304-307, 323-326 - the same format corrections as described above are needed

Response: Done.

We sincerely appreciate all valuable comments and suggestions, which helped us to improve the quality of the manuscript. 

Sincerely,

Authors

Reviewer 2 Report

My recommendation of References was ignored.

Author Response

Reviewer 2 comments

>> My recommendation of References was ignored.

#Response: We add the suggested references.

We sincerely appreciate all valuable comments and suggestions, which helped us to improve the quality of the manuscript. 

Sincerely,

Authors
